# Enhancement of zebrafish sperm production via a large body-sized surrogate with germ cell transplantation

Rigolin Nayak [1✉], Roman Franěk[1,2], Radek Šindelka[3] & Martin Pšenička[1]

Zebrafish (*Danio rerio*) is a commonly-used vertebrate model species for many research areas. However, its low milt volume limits effective cryopreservation of sperm from a single individual and often precludes dividing a single semen sample to conduct multiple down-stream procedures such as genomic DNA/RNA extraction and in-vitro fertilization. Here, we apply germ stem cell transplantation to increase zebrafish sperm production in a closely related larger species from the same subfamily, giant danio *Devario aequipinnatus*. The endogenous germ cell of the host is depleted by dead-end morpholino antisense oligonu-cleotide. Histology of the sterile gonad and quantitative PCR of gonadal tissue reveals all sterile giant danio develop the male phenotype. Spermatogonial cells of Tg(ddx4:egfp) transgenic zebrafish are transplanted into sterile giant danio larvae, and 22% of recipients (germline chimera) produce donor-derived sperm at sexual maturation. The germline chi-mera produce approximately three-fold the volume of sperm and 10-fold the spermatozoon concentration of the donor. The donor-derived sperm is functional and gives rise to viable progeny upon fertilization of donor oocytes. We show that the issue of low milt volume can be effectively addressed by employing a larger surrogate parent.

[1] The University of South Bohemia in Ceske Budejovice, Faculty of Fisheries and Protection of Waters, South Bohemian Research Center of Aquaculture and Biodiversity of Hydrocenoses, Vodnany, Czech Republic. [2] Department of Genetics, The Silberman Institute, The Hebrew University of Jerusalem, Jerusalem, Israel. [3] Laboratory of Gene Expression, Institute of Biotechnology, BIOCEV, Vestec, Czech Republic. ✉email: rnayak@frov.jcu.cz

Zebrafish (*Danio rerio*) is a popular model species used to study developmental biology, genetics, drug development, and many other research areas[1,2] because of their remarkable characteristics such as small size, easy breeding, external fertilization, transparent embryos with rapid development[3,4]. However, having many advantageous characteristics, zebrafish males produce only 1 μl of milt because of their small body size[5]. This can be a drawback for research requiring cryopreservation of sperm for genetic resource conservation of valuable zebrafish mutant strains and transgenic lines[6], particularly when maintaining the individual genotype precludes pooling sperm of several males. The motility and fertilization rate of thawed spermatozoa from such a small volume of semen may be extremely low, hampering the recovery of genetic lines[7–9]. Considering these limitations, we investigated the transplantation of zebrafish spermatogonial cells into one of their closely related species, the giant danio (*Devario aequipinnatus*), to assess its suitability as a surrogate parent for zebrafish. Giant danio belongs to the family *Cyprinidae* and is considered a phylogenetically closely related species to zebrafish[10]. Giant danio reaches 7–8 cm in captivity. It exhibits characteristics functionally similar to zebrafish. External fertilization and transparent eggs make it amenable to genetic manipulation, imaging, and visual observation. Notably, the fecundity of giant danio is greater than that of zebrafish. This species could potentially be a suitable surrogate for zebrafish to boost gamete production.

Surrogate broodstock production plays an important role in increasing aquaculture productivity[11]. The protocol requires transplantation of germline stem cells (GSC) from the donor of interest into a sterile host in order to produce donor-derived gametes. Success depends on factors such as (1) the evolutionary distance between the donor and recipient[12], (2) ablation of the host endogenous germ cell to prevent recipient-derived gamete production[13], and (3) a suitable developmental stage of the recipient[14,15]. The first attempt to produce germline chimeras in a fish was in zebrafish with the aim of determining if embryonic cells from the donor contributed to the host germline[16]. Germline stem cell transplant of a large-bodied species into a small fish with a shorter generation time has enabled rapid domestication of marine species, including Japanese yellowtail *Seriola quinqueradiata* transplanted into jack mackerel *Trachurus japonicus*[17] and commercially valuable tiger puffer *Takifugu rubripes* transplanted into grass puffer *Takifugu niphobles*[18]. Donor-derived viable sperm of landlocked Atlantic salmon *Salmo salar* was obtained with a triploid rainbow trout *Oncorhynchus mykiss* surrogate to reduce the generation time and improve environment adaptation[19]. Common carp *Cyprinus carpio* gametes were produced through spermatogonial cell transplantation into the peritoneal cavity of goldfish *Carassius auratus*[20] to preserve their germplasm in a smaller surrogate. Psenicka et al.[21] developed a strategy for germ cell cryopreservation of a nearly extinct sturgeon species. Such studies have shown that GSC transplantation tools can be a vital contribution to aquaculture.

Our study was designed to focus on the following key points; (1) to test the hypothesis if giant danio develops into male after endogenous germ cell ablation, (2) to enhance the fecundity of zebrafish using giant danio as a surrogate. Zebrafish is believed to have a polygenic sex determination system where multiple genes throughout the genome are identified as associated with sex determination[22]. Many studies have shown that zebrafish develop male-like characteristics after endogenous germ cell depletion[23–25], making it impossible to obtain donor-derived oocytes through sterile zebrafish surrogates by spermatogonial cell transplantation into the larvae. So, this study evaluated whether, being closely related to zebrafish, sterile giant danios recipients will also give rise to monosex male populations. If sterile giant danio recipients do not give rise to monosex male populations, we may be able to produce zebrafish donor-derived oocytes in giant danio surrogates. Giant danio surrogacy may be of significance considering the need to preserve hundreds of mutant zebrafish strains, which is currently possible only by sperm cryopreservation.

This study shows that giant danio developed the male phenotype after endogenous germ cell depletion. Sterile giant danio larvae transplanted with Tg(ddx4:egfp) zebrafish spermatogonial cells produced a significantly higher volume of fertile donor-derived sperm upon sexual maturation than the donor.

## Results

**Recipient production**. According to the study design (Fig. 1), we first spawn giant danio broodstocks to establish sterile recipients. We obtained ~3000 eggs from a single giant danio female and ~6 μl milt from individual males by abdominal pressure for in-vitro fertilization. The survival of MO-treated recipients was 50% of that of the control group (Table 1) and was consistent among all females. After injection, the positive control group injected with gfpUTRnanos3 and the MO co-injected with gfpUTRnanos3 group were screened under a fluorescence for GFP expression in the primordial germ cells (PGC). PGCs are the precursors of germ cell lineage that arise at the marginal part of blastodisc at the blastula stage and migrate to the genital anlage during embryogenesis[26–30]. The positive control showed the presence of primordial germ cells (PGCs) in all injected embryos ($n = 50$) during the segmentation period, whereas the MO co-injected with gfpUTRnanos3 group exhibited no GFP expression at any developmental stage (Fig. 2a, b'), confirming that the designed MO successfully prevented PGC migration in all injected embryos ($n = 50$) with 100% efficiency. Both male and female controls showed well developed gonads with mature spermatozoa and oocytes (Fig. 2d, e"). The gonads of morphants were challenging to identify after the dissection and degutting. Tissue on both sides of the gas bladder was negligible under the stereo microscope, and only identifiable after Bouin's fixation (Fig. 2c). The morphology of the gonad looked similar to sterile zebrafish testis (Fig. 2c', c")[13]. A similar structure was observed in other morphants, implying that the giant danios are males without germ cells (Supplementary Fig. 1). The RT-qPCR data of gonadal tissue showed expression of *sox9a* (essential for testes development) similar to that of the male controls and lower expression of *cyp19a1a* mRNA (expressed preferentially in female) (Fig. 2f). Sanger's sequencing confirmed the sequence specificity of all genes (Supplementary Note 1). All amplified sequences showed >93% identity to the zebrafish sequence after nucleotide BLAST. *Vasa* expression was negligible in all the morphants with both RT-qPCR and immunohistochemistry (Fig. 2f–i").

**Transplant success; germline chimera reproduction**. Initially, the donor testicular cells were prepared without density gradient centrifugation, and the transplant efficacy was low. Recipients with GFP-positive cells were scarce after one week ($n = 12$, 12%), and those cells gradually disappeared. No germline chimera was produced. After this, Ficoll was used to enrich the testicular cell suspension and to separate the excess spermatozoa from the larger cells, such as spermatogonia. Transplanted recipients ($n = 100$) were checked immediately after transplant to ensure the presence of Tg(ddx4:egfp) spermatogonial cells, and we found GFP-positive cells in all transplanted individuals (Fig. 3a, a'). The surviving recipients ($n = 85$, 85%) were screened one week post-transplant (1wpt) to identify the GFP-positive recipients ($n = 35$, 41.2%), which were observed to have few GFP-positive cells (see Fig. 3b, b'). After one month, a cluster of GFP-positive cells near the gas bladder was observed in five dissected fish (see Fig. 3c–e'). The number was increased from 1wpt, implying that the

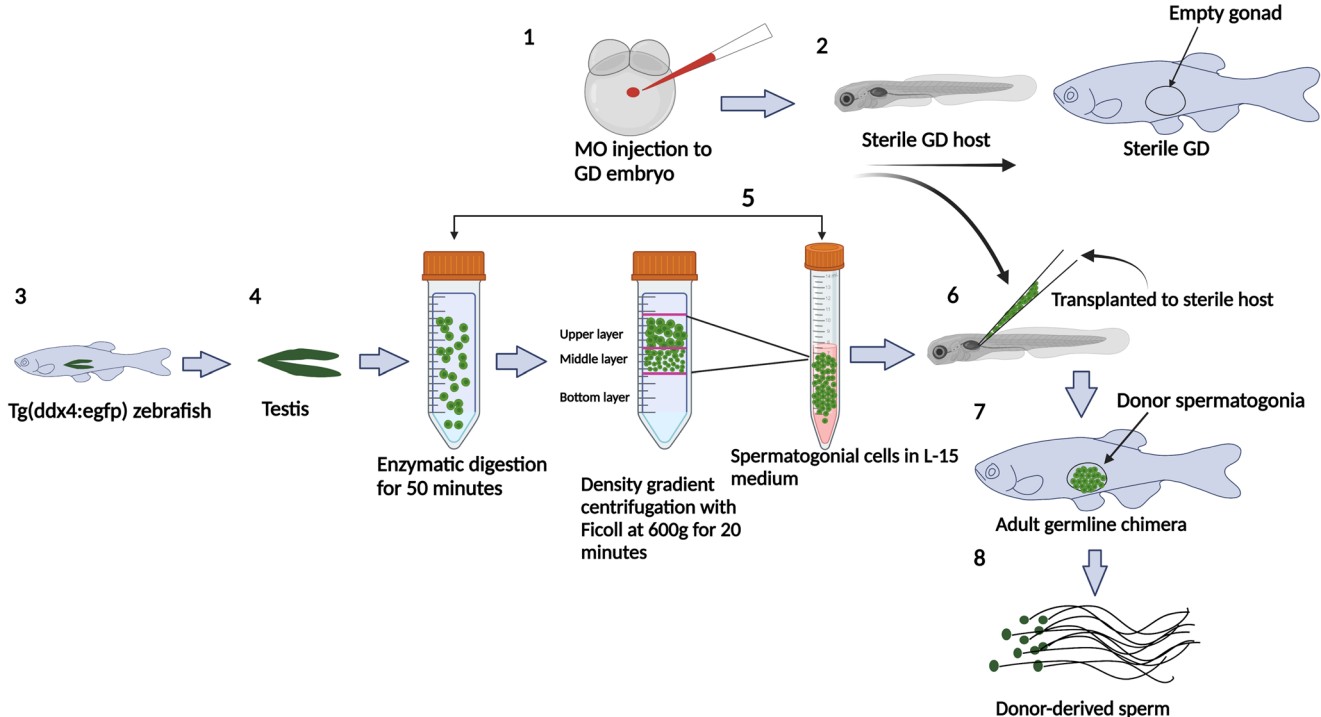

**Fig. 1 Schematic of study design.** GD-giant danio, MO-dead-end antisense morpholino oligonucleotide, L-15-Leibovitz's L-15 Medium. This figure was created with BioRender.com.

| Table 1 Survival rate of the recipients of morpholino injection. | | | | |
|---|---|---|---|---|
| | **Total embryos** | **25-somite (%)** | **Hatched (%)** | **Day-5 (%)** |
| Morpholino only | 257 | 193 (75.1%) | 145 (56.4%) | 123 (47.9%) |
| Control | 200 | 187 (93.5%) | 187 (93.5%) | 183 (91.5%) |

transplanted spermatogonia could incorporate into the giant danio empty gonad and proliferate. Three recipients died during the rearing process. The remaining recipients were stripped for milt collection at five months ($n = 27$). Only six males produced milt. Total sperm count ($1.28 \pm 0.4 \times 10^4$) and the milt volume were low (~1 µl) in all males, and the rate of fertilization of the donor egg was <20%.

A single milt-producing recipient was dissected to evaluate gonad development, revealing unilaterally developed testis with strong GFP expression. Histology showed a well-developed structure with all stages of spermatogenesis and the presence of mature spermatozoa (see Fig. 3f–h"), suggesting that the germline chimera is sexually mature at five months post-transplant, which is similar to the giant danio males. After a further month, the remaining 26 recipients were again stripped, with only the five surviving previously spawning males producing milt. GFP-specific PCR showed positive amplification in all five males (Fig. 3j), suggesting that the sperm produced was of donor origin (germline production rate 22%, $n = 6$). There was no GFP amplification in the giant danio control male. The sperm samples were examined under fluorescence microscopy to assess GFP expression, as the sperm from Tg(ddx4:egfp) lines also emits fluorescence because of residual protein near the spermatozoon head region. We observed the GFP signal in all spermatozoa (Fig. 3i, i').

**Reproductive characteristics of germline chimera.** A significantly higher volume of milt was collected from the five

| Table 2 Combinations of fertilization trials between the germline chimeras and the control group. | | |
|---|---|---|
| **Combinations** | **Female** | **Male** |
| ZFF × GCM | Zebrafish female | Germline chimera male |
| ZFF × ZFM | Zebrafish female | Zebrafish male |
| ZFF × GDM | Zebrafish female | Giant danio male |
| GDF × GDM | Giant danio female | Giant danio male |
| GDF × GCM | Giant danio female | Germline chimera male |

germline chimeras than from the zebrafish males but less than from the giant danio (Fig. 4a–c). A similar pattern was observed in spermatozoon concentration. Fertilization of donor oocytes was unsuccessful at the first spawning, likely due to the low number of spermatozoa. In the second in-vitro trial (Table 2), the rate of fertilization by the germline chimera males was similar to the zebrafish control groups, while the fertilization rate of the giant danio eggs by the germline chimera sperm was negligible (~5%). None survived to the segmentation period (Table 3). Information of the fertilization success of each germline chimera is presented in Supplementary Table 2. All swim-up (day 5) larvae (F1 progeny) produced by the donor-derived sperm were all positive for GFP-specific PCR amplifications, while the progeny from the AB control group showed no amplification. Few larvae from the AB control and germline chimera groups reared until their maturation. At two months post-fertilization, the adult progeny was examined under

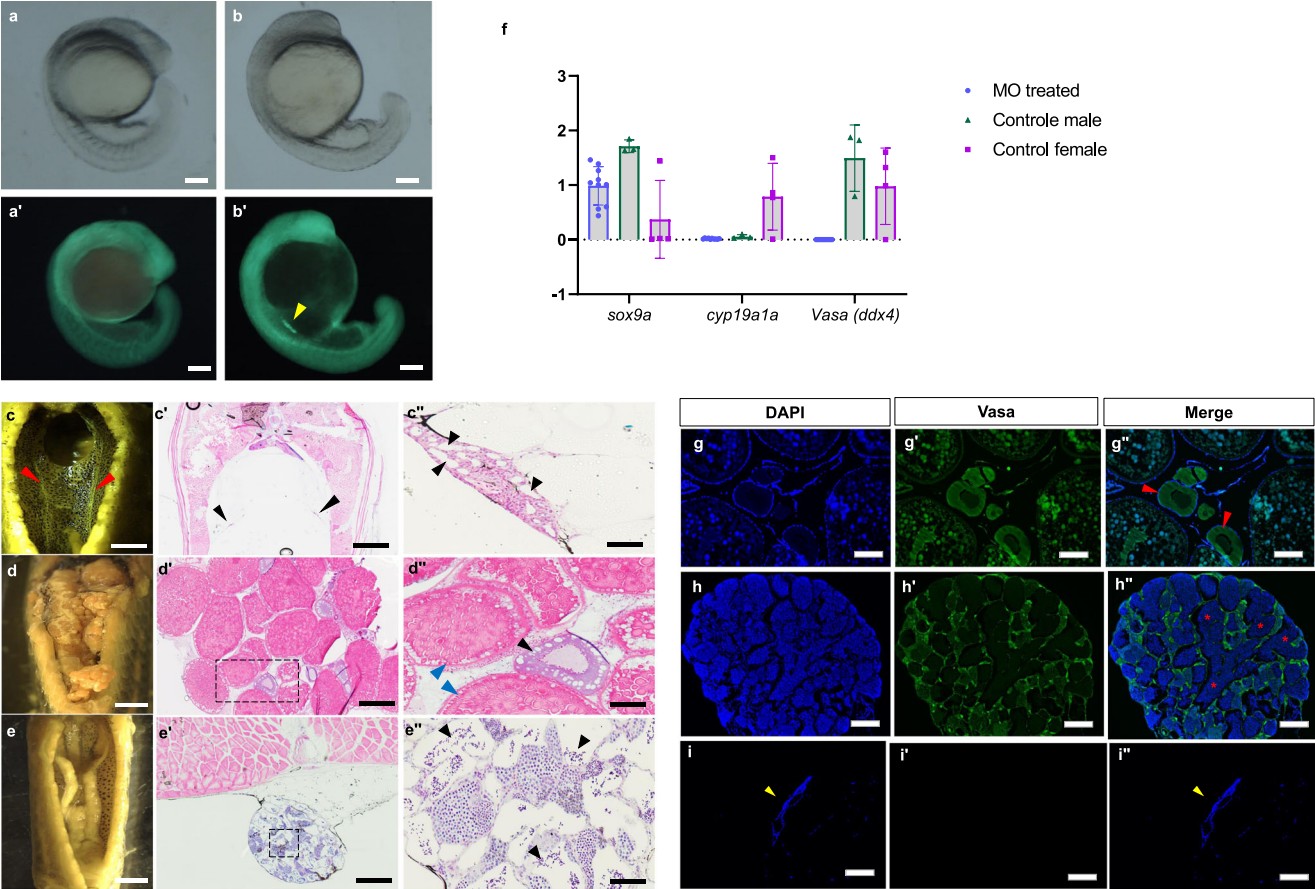

**Fig. 2 Dead-end knock-down of giant danio. a**, **a'** Embryo injected with MO plus gfpUTRnanos3 during the segmentation period shows no GFP-positive cells, $n = 50$. **b-b'** PGCs in a positive control injected with gfpUTRnanos3 only shows the presence of a cluster of PGCs under fluorescence (yellow arrowhead), $n = 50$, **c** The midsection of a 1-year-old morphant fixed with Bouin's shows a pair of underdeveloped gonads, thread-like structures tightly attached to the dorsal body wall (red arrowhead). **c'** Transverse section of the midsection and bilateral gonads (black arrowheads). **c"** Magnification of the region of the right gonad indicated by black arrowhead shows a testis-like structure with empty lumen lacking germ cells. A total of 10 gonads were analyzed for histology. **d**, **d"** Control giant danio female with well-developed ovary mostly filled with post-vitellogenin oocyte (blue arrowhead) and few cortical alveoli stages (black arrowhead). **e**, **e"** Control male with normally developed testis showing spermatozoa (black arrowhead). **f** The graph shows the expression levels of *sox9a*, *cyp19a1a*, and *Vasa* (ddx4) in the gonadal tissue of the morphants ($n = 10$ biological replicates), control males ($n = 3$ biological replicates) and control females ($n = 3$ biological replicates) relative to the expression level of reference gene *eef1a1l1*. The relative expression is calculated by the $2^{-\Delta\Delta Cq}$ method; data are presented as mean ± SD. Source data for this graph is provided in Supplementary Table 1. **g**, **g"** Sections of Vasa-Immunohistochemistry shows well-developed ovary of four-month-old control giant danio expressing prominent signal for Vasa (green) with red arrowheads indicating oocyte at the primary growth stage. **h**, **h"** transverse section of the control testis and red asterisks indicating the spermatozoa. **i**, **i"** Tissue sections of the MO-treated recipient (indicated by yellow arrowhead) with no expression of Vasa and signal only for DAPI. Three sterile recipients and three controls were analyzed for Vasa-Immunohistochemistry. Filter used, bright field for **a** and **b** and DA/FI/TRITC for **a'** and **b'**. Scale bars, **a**, **a'**, **b**, **b'**, **c"**, **d"** and **e"** = 100 µm, **c, d, e** = 1 mm, **c'**, **d'**, **e'** = 200 µm, **g**, **g"**, **h**, **h"**, **i** and **i"** = 100 µm.

fluorescence (Leica Fluorescence Microscope). We observed prominent GFP expression near the putative gonadal region in the progeny derived from the donor-derived sperm but none in the control individuals (Fig. 4d–e'). To confirm that the line of interest can be recovered, we applied a standard breeding protocol to obtain fish with the Tg(ddx4:egfp) transgene. The sexually mature F1 progeny were subjected to semi-in-vitro fertilization, and the resulting F2 progeny expressed a strong EGFP signal at different developmental stages (Fig. 4f–i'). The external appearance of the adult donor-derived progeny was similar to that of the zebrafish (Supplementary Figure 3).

## Discussion

We produced zebrafish donor-derived sperm with a total spermatozoon count of $67.02 \pm 27.39 \times 10^4$ (Mean ± SD, $n = 5$) and obtained viable progeny from giant danio surrogates by

spermatogonia transplant into the dead-end knock-down sterile host. In the preliminary experiment with total testicular cell suspension, we observed low survival of the transplanted cells at one week, decreasing to zero in all recipients. Thereafter, the donor testicular cell preparation was subjected to density gradient centrifugation to separate the spermatogonia from other cells (mainly spermatozoa). We used the same number of donors ($n = 4$) as in the previous protocol and observed improved transplant efficacy, suggesting that the excess number of spermatozoa in the cell suspension interferes with the spermatogonial cell incorporation into the giant danio gonad and affects their survival. We achieved 22% ($n = 6$) success with germline chimera production, and the mature individuals produced viable sperm and progeny when crossed with donor oocytes by in-vitro fertilization. The F1 progeny produced by the donor-derived sperm carried the Tg(ddx4:egfp) gene, confirmed by GFP-specific PCR of the larvae. The EGFP expression in the F1 progeny was only

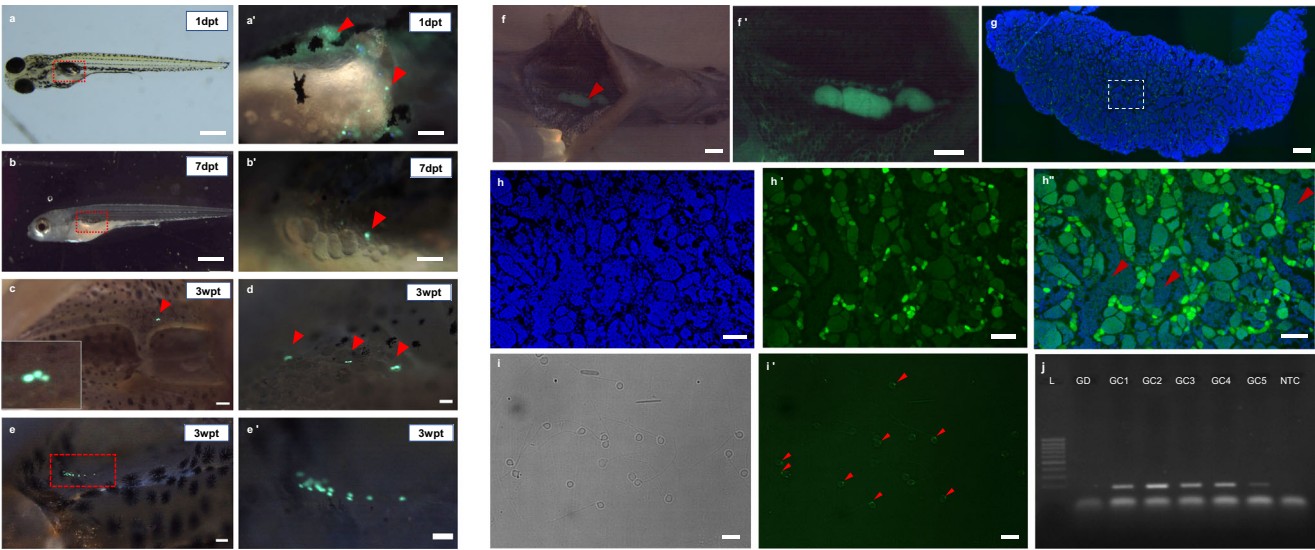

**Fig. 3 Intra-peritoneal spermatogonium transplant from Tg(ddx4:egfp) zebrafish into sterile giant danio recipients. a-a'** Detection of GFP-positive cells under fluorescence immediately after transplant into the posterior part of the gas bladder ($n = 100$). The magnified area is the red dashed rectangle, and cells are indicated by red arrowheads. **b, b'** The number of transplanted cells decreased seven days post-transplant, possibly because damaged spermatogonia and other cells, such as spermatids, are degraded over time, and few germ cells survive to proliferate. Eighty-five survived recipients were analyzed at 7dpt. **c, d** Two dissected recipients three weeks post-transplant (3wpt) show few GFP-positive cells in the genital ridge. **e** Another recipient of the same age group shows a cluster of cells indicated by a red dashed rectangle. Higher magnification **e'**. A total of five recipients were analyzed at 3wpt. **f** One five-month-old degutted germline chimera shows the presence of unilaterally developed testis under fluorescence microscope (FITC filter), and the red arrowhead indicates the testis. **f'** Higher magnification of the testis. **g** Longitudinal section of the same testis (merged image, the blue color is DAPI, and the green color is for GFP expression). **h** Higher magnification of the region depicted by the white rectangle from image **g**, under the DAPI filter, **h'** The same region under the FITC filer shows the expression of Vasa. **h''** Merge image for DAPI and FITC, and the red arrowheads indicate the spermatozoa populations. Approximately twenty sections were analyzed to check the Vasa expression **i, i'** Sperm from the germline chimera ($n = 5$) under brightfield and the GFP expression of the residual Vasa protein near the midpiece of the spermatozoa (indicated by red arrowheads) under the FITC filter. **j** The gel image of the five germline chimera (GC1-GC5) shows positive amplification for the GFP gene, and the product length is 187 bp loaded on 1.5% agarose gel. GD- sperm from giant danio males shows no PCR amplification, L-100bp DNA marker, and NTC is no template control. Uncropped and unedited gel image for this figure is provided as Supplementary Fig. 2. Filter used DA/Fl/TRITC (**a', b', c, d, e**, and **e'**) and brightfield (**a, b**). dpt – day post-transplant, wpt – week post-transplant. Scale bars: **a, b, f**, and **f'** = 1 mm, **b', a', c, d, e, e', h** and **h''** = 100 μm, **g** = 200 μm, **i** and **i'** = 10 μm.

observed in the adults because the Tg(ddx4:egfp) transgene is inherited from the male (Tg(ddx4:egfp) donor-derived sperm). Its expression was observed in the early developmental stages of the F2 progeny, since the Tg(ddx4:egfp) transgene was maternally contributed[31].

We attempted to boost milt production by giant danio surrogate and obtained higher sperm volume compared to zebrafish but than in the giant danio. The possible reason could be the development of one testis in the germline chimera, evident in post-transplant images. In all examined recipients, GFP-positive cells were observed on only one side of the coelomic cavity. A similar phenomenon of unilateral testis development was observed in the zebrafish after testicular cell transplant[13,32]. Nevertheless, the germline chimeras in this study produced more milt than did the donor. The spermatozoon concentration of the germline chimeras was higher than in the zebrafish, and the in-vitro fertilization rate was similar to that of the donor group. Giant danio males are reported to reach sexual maturation at five months and the zebrafish at ~3 months[33,34]. Our study showed that the germline chimera matured at five months, as evidenced by histology. However, the amount of milt was less, with a lower number of spermatozoa. Successful spawning was only obtained at six months, suggesting that the transplanted donor cells exhibit slower gonad maturation compared to giant danio males, delaying the onset of spermiation in the germline chimeras. This observation is not new, as few previous studies have reported zebrafish to become sexually mature ~ 6 months after germ cell transplantation[35,36]. We cannot completely exclude low germline

chimera production efficacy in this study. A possible reason for this could be the low abundance of Type A spermatogonia (SPG-A) in the transplanted cells. Previous studies have shown that only undifferentiated germ cells such as SPG-A have the potential to be incorporated into the host gonad[37,38]. Although we performed density gradient centrifugation for the spermatogonial isolation, it still could not ensure a higher proportion of SPG-A in the cell suspension. The enrichment of the SPG-A population by cell sorting may enhance the colonization efficiency in xenogenic transplantation[39].

Our study showed that giant danios develop as male after endogenous germ cell depletion. We randomly screened 10 dnd-MO-treated individuals and found a testis-like structure along with higher *sox9a* mRNA expression in the gonadal tissue and no *cyp19a1a* expression. Gonad-specific aromatase *cyp19a1a* is primarily involved in estrogen production[40], and estrogen stimulates ovarian development in fish[41]. The downregulation of *cyp19a1a* has been reported to promote masculinization[42]. On the contrary, *sox9a* is required for testis determination[43,44]. Moreover, no female germline chimera was found after transplantation, indicating that giant danio may require a certain number of PGCs to achieve the female fate, as is the case with medaka (*Oryzias latipes*) and zebrafish[23,25,45,46]. Xenogenic germ cell transplantation from other species to zebrafish host was performed[47,48] or between the different zebrafish strains[13,32]. We successfully established a zebrafish surrogate parent that can produce more sperm than the zebrafish. A significant increase in sperm volume and concentration in giant danio surrogates explains

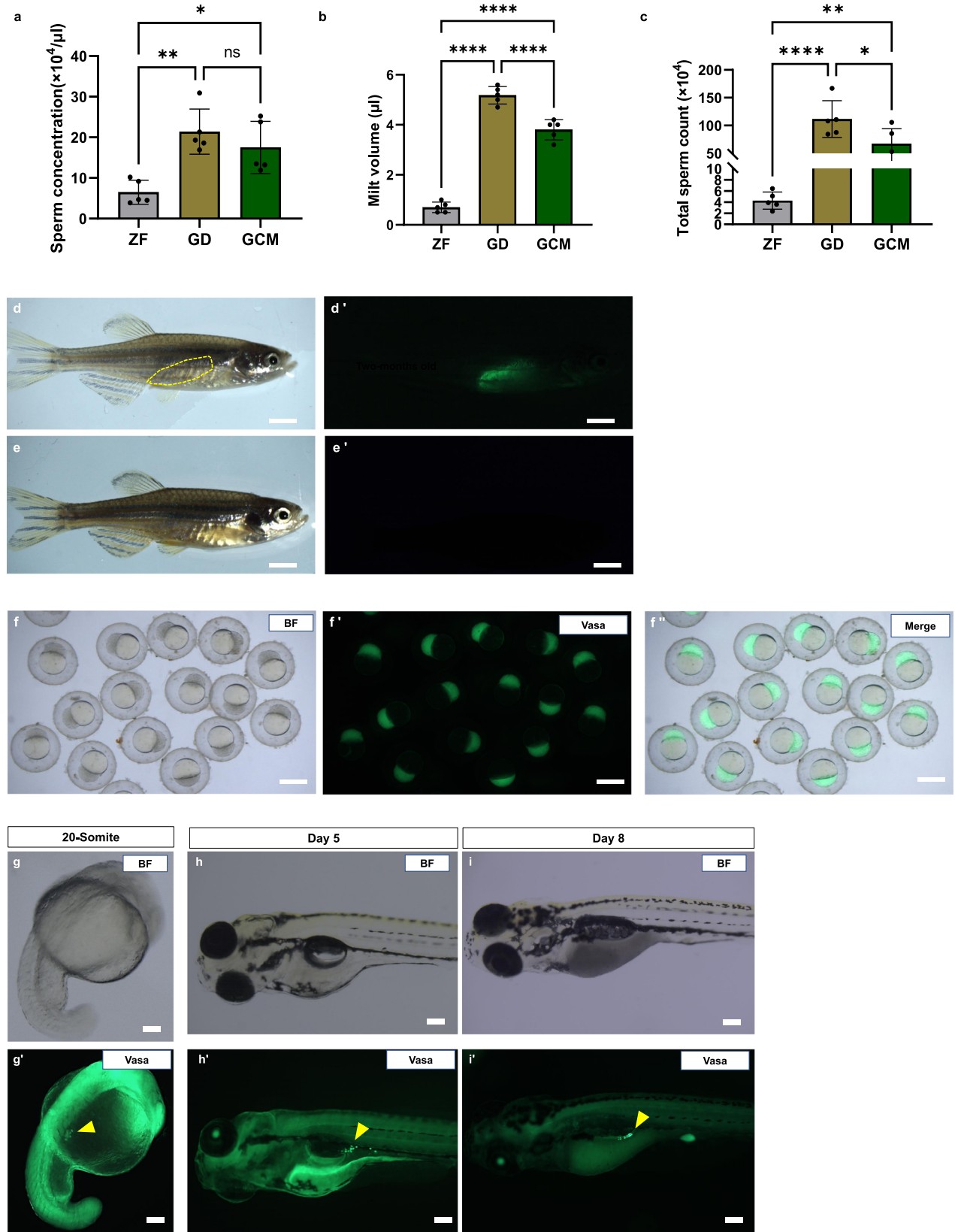

the relationship between body mass and reproductive output. Reportedly, reproduction output increases isometrically with body weight and size[49]. Gamete biomass scaled sub-linearly with body size in several species, including fish[50]. Giant danio is three times larger than zebrafish in size, which justifies the production of more milt volume in giant danio surrogates.

This represents the first report of zebrafish germ cell transplant into a host of a different genus. Although there are advanced methods developed for zebrafish sperm cryopreservation involving pooling the semen from many individuals[5] digestion of whole testes[8], it is challenging to cryopreserve the small volume of semen produced by an individual male[8]. Milt volume varies

**Fig. 4 Milt production of germline chimera and germline transmission in progeny from donor-derived sperm. a** Spermatozoon concentration expressed in ($10^4$ $\mu l^{-1}$), $n = 5$ biological replicates, **b** The total volume of milt obtained from three groups, $n = 5$ biological replicates, and **c** the total spermatozoon number, $n = 5$ biological replicates. Data are represented as mean ± SD. ZF, zebrafish males, GD, giant danio males and GCM, germline chimera males. $p < 0.0001$ (****), $p < 0.002$ (**), and $p < 0.0332$ (*). See Supplementary Table 3 for milt production of each germline chimera. **d** Two-month-old F1 progeny produced by donor-derived sperm and AB female oocyte under brightfield. The gonad region is delineated by the yellow border. **d′** The same fish under the FITC filter shows the expression of GFP in the designated area. **e** Control progeny under bright field. **e′**. Specimen showing no GFP expression under the FITC filter, both progeny were derived from oocytes of the same AB female. **f, f″** Embryo from the adult F1 progeny with strong GFP expression in the blastomere. **g, g′** F2 progeny at 20-somite stage showing GFP expression in PGCs, indicated by a yellow arrowhead. **h-h′** same embryo on day-5, and PGCs are indicated by a yellow arrowhead near the gas bladder. **i, i′** Eight days old larvae. Scale bars, **d, d′, e, e′** = 1 mm, and **f-i′** = 100 μm.

**Table 3 Fertilization success with donor-derived sperm. Data represent mean ± SD, $n = 5$ biological replicates; (-) indicates no survival in the group.**

| Group ($n = 5$) | Total eggs | 256-cell stage (%) | 25-somite (%) | Day 5 (%) |
|---|---|---|---|---|
| ZFF × GCM | 100 | 81.1 ± 4.5 | 76.7 ± 7.5 | 76.7 ± 7.4 |
| ZFF × ZFM | 100 | 84.7 ± 3.4 | 77.1 ± 5.4 | 75.7 ± 7.2 |
| ZFF × GDM | 100 | 4.6 ± 3.9 | — | — |
| GDF × GDM | 100 | 90.3 ± 5.7 | 87.5 ± 4.3 | 87.3 ± 4.6 |
| GDF × GCM | 100 | 4.8 ± 4.4 | — | — |

among zebrafish individuals, with some producing <1 μl. In our study, the germline chimeras produced sperm of ~10 fold the spermatozoon concentration of that of zebrafish, with maximum milt volume ranging from 3.2 μl to 4 μl (Supplementary Table 3). This may provide the potential to perform several downstream processes simultaneously from one individual. Many transgenic and other mutant zebrafish strains have been developed to study genetics, drug development, and other areas of human disease[51,52], and maintaining those valuable stocks cannot only be possible through sperm cryopreservation. Another method is required to overcome this issue. The germplasm of any zebrafish line can be transplanted into the giant danio and resurrected with more gametes since both giant danio and zebrafish take a similar period after transplantation to achieve sexual maturation. Giant danio culture and maintenance does not differ from that of zebrafish. Their feeding and breeding behavior are identical, and, like zebrafish, giant danio can reproduce multiple times per month. Sperm cryopreservation is not the only way to preserve the germplasm, and the confirmation of giant danio as a suitable surrogate parent for zebrafish means that studies may assess the potential for transplanting germ cells from a cryopreserved gonad into the giant danio recipients. Transplanted cryopreserved testes have been reported to colonize the zebrafish gonad and produce viable sperm[53]. A similar approach can possibly be applied to giant danio. There are many reports of germline chimera production via transplanting cells of larger species into small-bodied species in order to shorten the reproductive cycle[17,20]. Our study reports surrogate production techniques that can be helpful in increasing gamete production by xenogenic spermatogonial transplantation from smaller species into a larger recipient. Although there is existing literature reporting goldfish (*Carassius auratus*) gamete increment via common carp (*Cyprinus carpio*) surrogates, where common carp recipients were treated with busulfan (myleran) for endogenous germ cell ablation[54]. However, we cannot exclude the fact that busulfan cannot ensure the complete depletion of all the germ cells in the gonad and may give rise to recipient-derived gametes[11]. Higher dosage of busulfan also causes cytotoxicity and increase mortality in the treated individuals[55]. Dead-end gene knockdown by antisense morpholino oligonucleotide which is applied in our study for recipient production has been reported to be more effective than any other method for germ cell depletion for surrogate production in fish[13].

In summary, our study established a surrogate parent for zebrafish from a different genus with highly concentrated donor-derived sperm and a fertilization rate comparable to that of the donor individuals. Giant danio germline chimeras may provide a solution to the limitations encountered with low-sperm production by zebrafish individuals and potentially be used to preserve the transgenic and mutant zebrafish lines produced to use for various research purposes.

## Methods

**Fish husbandry.** Giant danio broodstock were maintained in the Faculty of Fisheries and Protection of Waters (FFPW), the University of South Bohemia in Ceske Budejovice, Vodňany. Rearing temperature was $25 ± 0.5$ °C and photoperiod 14 L:10D. Fish were fed commercial dry flakes twice, and blood worms once, daily. Zebrafish lines were obtained from the University of Liege, Belgium[32], held in our facility for several generations at 28 °C, photoperiod 14 L:10D, and fed *Artemia nauplii* and dry diet. All the experiments were conducted at the facility of FFPW.

**Dead-end (*dnd*) gene amplification and morpholino designing.** Total RNA was extracted from the gonads of three-month-old giant danio using the RNeasy Mini Kit (Qiagen). Extracted RNA was processed for cDNA synthesis using WizScript™ RT FDmix, thermal cycler programmed at 25 °C for 10 min, 42 °C for 30 min, 85 °C for 5 min, and held at 4 °C. The *dnd* gene was amplified from the cDNA using 0.5 μl of forward primer 5′-TCCACCAATTTACAGGTGTGTC-3′ and reverse primer 5′-CGAGGCTGTAAGAGGGTCAC -3′ and 5 μl of PPP master mix (Top Bio). The primers for the *dnd* gene were designed from the available zebrafish *dnd1* mRNA sequence (accession number NM_212795.1). The amplification conditions were 94 °C for 5 min and 35 cycles of 94 °C for 30 s, 58 °C for 30 s, and 72 °C for 45 s with a final extension of 72 °C for 5 min. The amplified product was checked for positive amplification on 2% agarose gel (Supplementary Fig. 4). The amplicon was cut with a sterile scalpel from the gel and purified using the GeneAll ExpinTM Combo GP kit, followed by Sanger's sequencing. Three individuals of each sex were sequenced to confirm the sequence specificity. The dnd-morpholino (MO) was designed according to the start codon of the cDNA sequence (Supplementary Note 1 shows the sequence of the *dnd* amplicon). The resulting *dnd* sequence was compared with available *dnd* sequences (data obtained from the NCBI database) from other species including zebrafish (Accession number NM_212795.1), medaka (NM_001164516.1), goldfish (JN578697.1), common carp (MN447719.1), and rainbow trout (NM_001124661.1) to identify conserved nucleotides across species (Supplementary Fig. 5). The designed MO sequence for giant danio is 5′-CTGTAAATGCCGTTGAGCCTCCATG-3′.

**Sterile recipient production.** Giant danio broodstock were held in breeding chambers (6.8″ L × 4″ W × 3.9″ H) at $25 ± 0.5$ °C. Males and females were separated by a barrier in the afternoon prior to spawning. The following day, barriers were removed, allowing mating, and fish were observed for oviposition. Gametes were collected from breeding pairs: Semen from the males was collected in E400 extender (immobilization solution at 1:10 dilution) by gentle abdominal massage. Eggs collected from the ovulating females into a clean Petri dish, by gentle abdominal pressure, were promptly fertilized in vitro. Post-fertilization embryos were divided into four groups: (1) MO only ($n = 257$), (2) MO co-injected with gfpUTRnanos3 (to confirm the depletion of PGC, $n = 50$)[56], (3) gfpUTRnanos3 only (positive control to confirm that PGCs are labeled, $n = 50$), and (4) intact control ($n = 200$). 1 mM stock MO was diluted to 0.1 mM with nuclease-free water (Ambion™, ThermofisherScientific) and 2 M potassium chloride to a final concentration 0.2 M. Embryos were injected through the chorion at the 1–2 cell stage into the yolk near the blastodisc under a stereomicroscope using a glass capillary (World Precision Instruments, Item No. 1B100-4) connected to a micromanipulator (M-152, Narishige, Japan) and FemtoJet 4× microinjector (Eppendorf, Germany). All experimental groups were cultured at $25 ± 0.5$ °C with a daily change

of fresh temperate water. The survival rate of MO-treated embryos and intact controls were recorded until the day 5.

**Transplantation**. Three-month-old Tg(ddx4:egfp) male zebrafish ($n = 4$) were killed with overdose of anesthetic tricaine methane sulfonate (MS222). Testes were aseptically removed, placed in 1× phosphate-buffered saline (PBS), and cut with sterile scissors into small fragments then washed twice to remove excess sperm. The tissue fragments were transferred into the cell dissociation solution containing PBS with 0.1% trypsin, 0.05% collagenase (Gibco), and 30 μg/ml of DNase. The tissue was further minced with scissors and incubated for 50 min at 22–23 °C on a rocking shaker. The cell suspension was then mixed with L-15 (Leibovitz medium) cell culture medium supplemented with 10% (final concentration) fetal bovine serum (FBS) at a 1:1 ratio and filtered through a 30 μm pore (CellTrics) to remove debris. The cell suspension was subjected to density gradient centrifugation using Ficoll-Paque ™ PLUS (GE Healthcare) at $600 \times g$ for 20 min. After centrifugation, two upper layers containing early-stage spermatogonia were carefully transferred to a new sterile tube according to Panda et al.[57]. The collected cells were washed with 1× PBS to remove ficoll at the same speed, and the cell pellet was resuspended with the 30 μl of L-15 medium with 10% FBS. Seven-day-old germ cell–depleted recipients were anesthetized with 0.08% MS222 and placed on the Petri plate coated with 1% agar. The spermatogonia cell suspension was loaded into the pulled glass capillary connected to the injector as described above for morpholino injection. Approximately 500–600 cells were transplanted into the posterior part of the gas bladder in the recipients ($n = 100$).

**Germline chimera reproduction and donor-derived gamete confirmation**. One-week post-transplant (1wpt), recipients ($n = 100$) were anesthetized with 0.08% MS222 and checked for the presence of GFP-positive cells under fluorescence microscopy (Leica Microsystems). Individuals with GFP-positive cells were reared separately in the incubator and fed with paramecium for two weeks, followed by *Artemia nauplii* until two months, then transferred to the aquarium and fed with dry flakes two times and blood worms once daily. Four weeks post-transplant, five fish were dissected to detect whether transplanted cells colonized the recipient gonad. Five months post-transplant, semen samples were collected from the mature recipients by stripping. Genomic DNA was extracted from the semen samples by the hotshot DNA extraction method[22], followed by PCR amplification using GFP-specific primers, forward 5′- ACGTAAACGGCCACAAGTTC -3′, and reverse 5′- AAGTCGTGCTGCTTCATG -3′[32]. Semen of a control giant danio was used as negative control for chimera genotyping.

**In-vitro fertilization and spermatozoon concentration assessment**. Zebrafish AB strain males and females were separately kept in a breeding chamber with the help of a barrier a day before spawning for fertilization. The same method was applied for giant danio males and females. On the spawning day, barriers were removed, and the females were observed for oviposition. Breeding pairs were first anesthetized with 0.08% MS222. Eggs from 7 to 8 zebrafish ovulated females were collected together in one Petri dish by gentle abdominal pressure. Similarly, two ovulated giant danio females were stripped to obtain the eggs. Semen samples from the control males and the germline chimera males were collected separately in the E400 extender (10 μl of the extender was used for each 1 μl of milt). A total of 10–20 μl of diluted sperm was used to fertilize ~100 eggs for each group (0.2–0.5 ml of dechlorinated water was used for sperm activation). Several combinations were made (see Table 2) in five replicates (five independent biological replicates) for the fertilization trial. The fertilization and the survival rate were recorded until day 5 for each combination, and the larvae derived from donor-derived sperm were subjected to GFP-specific amplification to detect germline transmission. A part of the progeny was reared until maturation and reproduced to generate F2 progeny.

Next, the semen samples were further diluted (1:10) to determine the sperm concentration for the donor-derived semen samples ($n = 5$) and controls ($n = 3$, for both donor and host). A Bürker Counting Chamber was used for spermatozoa counting.

**Histology**. The gutted torsos of 8-month-old morphant ($n = 10$) and control recipients were fixed in Bouin's fixative for 24 h and dehydrated in an ethanol series, embedded in JB-4 resin (JB4 embedding kit), using a plastic mold[58], and cut into 5 μm sections with a rotary microtome (Leica Biosystems). Sections were stained with hematoxylin and eosin and examined under fluorescence microscopy (Olympus BX51).

A five-month-old germline chimera was dissected, and the entire testis was fixed with 4% PFA for 3 h at 4 °C and washed with 1× PBS. The fixed tissue was dehydrated with ethanol dilutions to preserve the GFP fluorescence (10 min for each dilution at 4 °C) and embedded in JB-4 resin. Twelve hours post-embedding, the tissue was sectioned and placed onto the SuperFrost slides with 5 min of air drying and mounted with fluoroshield mounting medium with DAPI. The prepared slides were analyzed by fluorescence microscopy immediately.

**Sex-specific gene expression of sterile giant danio by RT-qPCR**. Total RNA was extracted using TriReagent extraction and LiCl precipitation according to the manufacturer's instructions. The concentration of total RNA was determined using a spectrophotometer (Nanodrop 2000, ThermoFisher Scientific), and the quality of RNA was assessed using a fragment analyzer (AATI, Standard Sensitivity RNA analysis kit, DNF-471). The cDNA was prepared using 500 ng of total RNA, 0.5 μl oligo dT and random hexamers (50 μM each), 0.5 μl dNTPs (10 mM each), 0.5 μl Maxima H Minus Reverse Transcriptase (Thermo Scientific), 0.5 μl recombinant ribonuclease inhibitor (RNaseOUT, Invitrogen), and 3 μl 5× Maxima RT buffer (Thermo Scientific) mixed with Ultrapure water (Invitrogen) to a final volume 15 μl. Samples were incubated for 5 min at 65 °C, 10 min at 4 °C, 10 min at 25 °C, 30 min at 50 °C, and 5 min at 85 °C followed by cooling to 4 °C. Obtained cDNA samples were diluted to a final volume of 100 μl and stored at -20 °C.

The qPCR reaction contained 5 μl of TATAA SYBR Grand Master Mix, 0.5 μl of forward and reverse primers mix (mixture 1:1, 10 μl each), 2 μl of cDNA, and 2.5 μl of RNase-free water. The reaction was conducted using the CFX384 Real-Time system (BioRad) with initial denaturation at 95 °C for 3 min followed by 40 cycles of denaturation at 95 °C for 10 s, annealing at 60 °C for 20 s, and elongation at 72 °C for 20 s. Melting curve analysis was performed to test reaction specificity, and only one product was detected for all assays. Male and female gonadal tissue was processed similarly to validate the specificity of RT-qPCR detection, and PCR products (50 ng) from MO samples were analyzed using standard Sanger sequencing with forward and reverse gene primers to confirm amplification specificity. The qPCR primers were designed from the available zebrafish mRNA sequences *sox9a* (Accession No. NM_131643.1), *cyp19a1a* (Accession No. NM_131154.3), *Vasa* (Accession No NM_131057.1), and reference gene eukaryotic translation elongation factor 1 alpha 1, like 1 *(eef1a1l1)* (Accession No. NM_131263.1). For the primer list, see Supplementary Table 4.

**Vasa-immunohistochemistry of the sterile giant danio gonad**. Six-month-old giant danio morphants ($n = 3$) and controls were over-anesthetized, as mentioned. The gonads were removed and fixed with 4% PFA overnight at 4 °C, followed by dehydration and paraffin embedding. Sections were deparaffinized with xylene and with 96% and 70% ethanol, followed by washing twice with distilled water and PBS. The sections were subjected to antigen retrieval with sodium citrate buffer at 96 °C for 10 min, permeabilized with 0.3% triton in PBS for 10 min, and blocked with 10% goat serum. The sections were incubated with primary antibody Anti-DDX4 (ab13840) diluted to 1:300 with 4% goat serum in PBS overnight, followed by secondary antibody (goat anti-rabbit IgG (H + L) Cross-Adsorbed Secondary Antibody, Alexa Fluor™ 488) at 1:500 dilution with PBS overnight, mounted with fluoroshield with DAPI and examined by fluorescence microscopy.

**Statistics and reproducibility**. Data on spermatozoon concentration, milt volume, and total sperm count were first checked for normal distribution using the Shapiro-Wilk test. Differences among three groups were calculated using one-way ANOVA followed by Tukey's multiple comparisons with the adjusted $p < 0.05$, with data expressed as mean ± SD. For each group, data were collected from five biological replicates and repeated an average of three times. $p < 0.0001$ (****), $p < 0.002$ (**), and $p < 0.0332$ (*) unless stated otherwise. Sample size was determined based on the surviving germline chimera. No data was excluded. All statistical analysis was performed using GraphPad prism 9. Fertilization success of donor-derived sperm from germline chimera and control groups is presented as mean ± SD; data for each group was derived from five biological replicates (each biological replicate represents one independent fish). In-vitro fertilization was repeated three times for all the combinations. RT-qPCR data were derived from the gonadal tissue of ten MO-treated fish, three control males, and three control females. RT-qPCR was independently repeated two times to check the consistency of the results.

**Reporting summary**. Further information on research design is available in the Nature Portfolio Reporting Summary linked to this article.

## Data availability

The source data for Fig. 2f is provided as Supplementary Table 1, and Fig. 4a–c is provided as Supplementary Table 3. The uncropped and unedited gel image for Fig. 3j is provided as Supplementary Fig. 2. All data generated or analyzed during this study are included in this article and its Supplementary Information. All other data are available from the corresponding author on reasonable request.

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

## Acknowledgements

The work was supported by the Ministry of Education, Youth and Sports of the Czech Republic, project Biodiversity (CZ.02.1.01/0.0/0.0/16_025/0007370), and the Czech Science Foundation (grant number 22-31141 J and grant to RF 22-01781 O). The project received funding from the European Union's Horizon 2020 Research and Innovation Programme under grant agreement No 871108 (AQUAEXCEL3.0). This output reflects only the authors' views, and the European Union cannot be held responsible for any use that may be made of the information contained therein.

## Author contributions

M.P. conceptualized and supervised the study, R.F. performed immunohistochemistry, R.S. performed qPCR and Sanger's sequencing for qPCR products. R.N. designed the study, performed the transplantation and other experiments, and wrote the manuscript. All authors helped in the manuscript editing and approved the final version.

## Ethics declarations

The study was conducted at the Faculty of Fisheries and Protection of Waters (FFPW), the University of South Bohemia in Ceske Budejovice, Vodnany, Czech Republic. The facility has the competence to perform experiments on animals (Act no. 246/1992 Coll., ref. number 16OZ19179/2016–17214). The Institutional Animal Care and Use Committee of the FFPW approved the methodological protocol of the current study according to the law on the protection of animals against cruelty (reference number: MSMT-6406/2019–2). The study did not involve endangered or protected species. Authors of the study (R.F. and M.P.) possess the Certificate of Professional Competence for designing experiments and experimental projects under Section 15d (3) of Act no. 246/1992 Coll. on the Protection of Animals Against Cruelty.

## Competing interests

The authors declare no competing interests.
