## [Peer Review File · Communications Biology]

Reviewers' comments:

Reviewer #1 (Remarks to the Author):

In the submitted manuscript, Nayak et al. transplanted centrifuge-isolated spermatogenic cells of zebrafish into sterile giant danio larval fish, and obtained zebrafish sperm with high volume and high concentration of spermatozoa. The Figure 1 describes the design of this study, the Figure 2-3 describe the production of sterile host fish (giant danio), and the Figure 4-5 describe the generation of germline chimeras by germ cell transplantation, and the last two figures describe the production of zebrafish sperm by giant danio. Although the whole study design is straightforward and results are mostly reliable, some major concerns or changes should be solved before further consideration.

Major points:

1. Figures: Figure 2 and Figure 3 should be combined, Figure 4-5 should be combined, and Figure 6 and Figure 7 should be combined. Figure 2a is with low resolution, it could be enlarged and moved to supplemental data.
2. Statistic issues: Most figures lack statistical analysis, such as the efficiency of PGC depletion in Figure 2 b-c, how many sections/gonads were analyzed for Vasa-immunostaining in Figure 3c-c'', how many fish were analyzed in Figure 4 a-e' and Figure 5a-a', and how many sections were analyzed in Figure 5b-d'. Although some statistical data were present in the main text, it is necessary to put all the data in these figure panels.
3. Although there was a maternal effect for the GFP expression in the Tg(vasa:gfp) transgenic fish, they should present germline-specific expression of GFP at later developmental stages ([https://doi.org/10.1016/S0925-4773\(02\)00154-5](https://doi.org/10.1016/S0925-4773(02)00154-5)). The authors should show the germline expression of the fish derived from the surrogate F1 progeny.
4. The authors designed and utilized a dead end morpholino (MO) to block the translation of endogenous dead end in giant danio, but they did not follow a guideline to prove the effectiveness and specificity of this MO, even by using a GFP-fused mRNA approach as described previously (<https://doi.org/10.1371/journal.pgen.1007000>, <https://doi.org/10.1002/adv.202203631>).
5. This manuscript also missed some highly relative literatures, such as zebrafish were used as surrogate host to produce sperm of other species from another subfamily by spermatogonial cell transplantation (<https://doi.org/10.1007/s11427-021-1989-9>).
6. All the transgenic fish and gene name should follow the standard nomenclature of zebrafish, such as Tg(ddx4:egfp), instead of vas::EGFP in the manuscript, gfpUTRnanos3, instead of GFP-nos1-3'UTR in the manuscript.

Minor points:

1. Line 29, should clearly define the genetic relationship between zebrafish and giant danio.
2. Line 59, it is not clear whether they used germ stem cells for transplantation, thus it is not appropriate to use the phrase "germ stem cells" or "spermatogonial stem cells" in this study. Suggested to use the phrase "spermatogonial cells" in the whole manuscript.
3. Line 63, should give a reference for the maturation period of giant danio. Otherwise, this description should be moved to the Results section.

4. Line 66-68, should provide reference for this sentence. Otherwise, this description should be moved to the Results section.
5. P70, an extra dot.
6. P86, are > were.
7. Line 97-98, this conclusion is not accurate. It is possible to obtain donor-derived oocytes when zebrafish PGCs were transplanted into germ cell-depleted host embryos (<https://doi.org/10.1073/pnas.222459999>, <https://doi.org/10.1016/j.jgg.2019.12.004>).
8. Figure 1, it is not accurate to use the phrase “spermatogonial stem cells”, and it is also needed to clearly show how to obtain the donor cells for transplantation.
9. Line 114, there should be a reference for the Tg(ddx4:egfp) transgenic line.
10. Line 123-125, it is not clear why the primers for amplifying giant danio dnd gene was designed according to the zebrafish dnd1 mRNA sequence.
11. Line 139-140, there should be an approach to clearly define the specificity and effectiveness of the newly designed MO.
12. Line 150, GFP-nos1-3'UTR should be gfpUTRnanos3.
13. Line 160, swim-up stage should be standard stage name, according to https://zfin.org/zf_info/zfbook/stages/index.html. There is an error for the reference.
14. Line 163, delete the word “fish”.
15. Line 254, please provide the standard name of the gene “elongation factor”.
16. Line 282, define PGCs here.
17. Line 284, lack statistic data. There is an error for the reference,
18. Figure 2g, please compare the expression levels of each gene between different samples, but not different genes in a certain sample.
19. Line 322, please use the standard stage names for these stages in Table 1.
20. Line 325, please remove all the “AB” in Table 2. It is unnecessary to mention AB in the Table.
21. Line 357, Figure 3, please label the genotype or injection condition in the panels.
22. Line 396-308, needs to verify the germline specific EGFP signal in the F2 progeny.
23. Line 401, Table 3, how many replicates have been conducted to test the fertilization rate? How could the authors obtain exactly 100 eggs in each group?
24. Line 426, 104 µl should be corrected.
25. Line 433, Figure 7, the author should check the germline-specific expression of EGFP in the F2 progeny.
26. Line 461-462, it is true that EGFP has maternal contribution, but the authors need to check the germline expression at later stages.
27. Line 477-478, it is not necessarily true. In previous studies, germ cell transplanted zebrafish also reached sexual maturation at 3 months ().
28. Line 480, ASG should be SPG-A.
29. Line 490, cyp191a should be cyp19a1a
30. Line 491, secretion should be production.
31. Line 496, a reference is missing (<https://doi.org/10.1007/s10126-019-09874-1>).
32. Line 497, spermatogonial cell transplantation from another species into zebrafish was also achieved (<https://doi.org/10.1007/s11427-021-1989-9>)

Reviewer #2 (Remarks to the Author):

Zebrafish is a useful experimental model, but its low semen volume makes sperm cryopreservation difficult. The authors have succeeded in producing sperm derived from a donor zebrafish from 6-month-old recipients by preparing germline stem cells from the testis of a 3-month-old zebrafish and transplanting them into the recipients. The authors estimate that the amount of sperm produced by the recipient was 10 times more than that of normal zebrafish. Although this idea is very interesting, this reviewer did not find sufficient merit in this technique for the following two reasons.

(1) Zebrafish biobanking usually involves mincing the testes to obtain a large amount of sperm, making it possible to obtain a sufficient amount of sample for cryopreservation.

(2) In this study, 3-month-old individuals were used as donors, and a large amount of zebrafish sperm was obtained 6 months after the germ cell transplantation. With this amount of time, two more generations of zebrafish can be mated, which can increase the number of sperm-producing individuals by several hundred times.

In addition, this phenomenon is not new from a biological point of view, since germ cell transplantation using zebrafish as donors has already been reported in several papers, and interspecific germ cell transplantation within the same genus has also been reported successfully in several fish species.

Reviewer #3 (Remarks to the Author):

General comment:

This manuscript reported the germ-line chimeras from zebrafish as donor to giant danio as host, in which sperm derived from donor were produced. In the germ-line chimera, volume and concentration of sperm were higher than those of donor species. As mentioned in the manuscript by the authors, this finding would be able to contribute several applications such as sperm cryopreservation. From another view point, this result is interesting to study the definition of gonadal size in species. But the authors did not mention the size of testes of germ-line chimeras and compared size of testes among germ-line chimera and species used in this study. Furthermore, it might be possible to discuss the ability of proliferation in germ cells derived from donor species. I wonder if the authors could add the information about the size of their gonads and discuss the gonadal development of germ-line chimera to improve this manuscript for basic biological interests.

Specific comments:

Line 160. No need to cite Table 1 in this part.

Line 199-210. How did you conduct artificial fertilization? How many eggs and how much volume of diluted semen were used for the fertilization trials in this study, and then how did you activate them for

fertilization?

Line 350-352. Figure 5e should be cited.

Line 353-356. In figure 5d-d', the authors indicated four spermatozoa by arrowheads. Could you detect GFP expression in all sperm observed.

Point-by-point response to the referees

Reviewers' comments:

Reviewer #1 (Remarks to the Author):

In the submitted manuscript, Nayak et al. transplanted centrifuge-isolated spermatogenetic cells of zebrafish into sterile giant danio larval fish and obtained zebrafish sperm with high volume and high concentration of spermatozoa. The Figure 1 describes the design of this study, the Figure 2-3 describe the production of sterile host fish (giant danio), and the Figure 4-5 describe the generation of germline chimeras by germ cell transplantation, and the last two figures describe the production of zebrafish sperm by giant danio. Although the whole study design is straightforward and results are mostly reliable, some major concerns or changes should be solved before further consideration.

We thank the reviewer for the positive remarks and a thorough review of the manuscript.

Major points:

1. Figures: Figure 2 and Figure 3 should be combined, Figure 4-5 should be combined, and Figure 6 and Figure 7 should be combined. Figure 2a is with low resolution, it could be enlarged and moved to supplemental data.

- Figure 2 and 3 are combined and provided as figure 2.
- Figure 4 and 5 are combined and provided as figure 3.
- Figure 6 and 7 are provided as figure 4.
- Figure 2a is moved to supplementary file 1 as Figure 2.

2. Statistic issues: Most figures lack statistical analysis, such as the efficiency of PGC depletion in Figure 2 b-c, how many sections/gonads were analyzed for Vasa-immunostaining in Figure 3c-c'', how many fish were analyzed in Figure 4 a-e' and Figure 5a-a', and how many sections were analyzed in Figure 5b-d'. Although some statistical data were present in the main text, it is necessary to put all the data in these figure panels.

A total of 50 embryos were injected with MO plus gfpUTRnanos3 (treated group) and 50 with gfpUTRnanos3 only (Positive control). All 50 embryos from the positive control group showed the presence of PGCs with strong GFP expression, and none of the embryos from the treated group showed GFP expression for PGCs. The data for efficiency is now updated in the result section 3.1 and the figure legend (figure 2a-b').

Three sterile recipients and three male and female controls were analyzed for Vasa-immunohistochemistry, and the numbers are now inserted in the figure legend (figure 2g-l').

All 100 recipients were checked for the presence of Tg(*ddx4:egfp*) spermatogonial cells immediately after transplantation (Figure 3a-a'). After one week, 85 surviving recipients were analysed (figure 3b-b'), and five recipients were dissected at 3 weeks post-transplant (Figure 3c-e'). One five-month-old germline chimera was sacrificed for vasa expression (figure 3 f-f'), and Approximately twenty sections were analysed under fluorescence (figure 3g-h') from the gonad shown in figure (figure 3 f-f'), and we found vasa expression in all the sections analysed. Sperm samples from all five germline chimera were analysed under fluorescence to check the Vasa expression

All data are now present in the respective figure panels.

3. Although there was a maternal effect for the GFP expression in the Tg(*vasa:gfp*) transgenic fish, they should present germline-specific expression of GFP at later developmental stages ([https://doi.org/10.1016/S0925-4773\(02\)00154-5](https://doi.org/10.1016/S0925-4773(02)00154-5)). The authors should show the germline expression of the fish derived from the surrogate F1 progeny.

We apologize for not providing the images of F2 progeny at later developmental stages. Now we have added new figures of F2 progeny at 20-somite, day five, and day eight stages in figure 4.

4. The authors designed and utilized a dead end morpholino (MO) to block the translation of endogenous dead end in giant danio, but they did not follow a guideline to prove the

effectiveness and specificity of this MO, even by using a GFP-fused mRNA approach as described previously (<https://doi.org/10.1371/journal.pgen.1007000>, <https://doi.org/10.1002/adv.202203631>).

We do apologize for not following the provided guidelines. On the other hand we would like to emphasize that purpose of the presented manuscript was not to investigate the effect of dnd MO itself on giant danio since we used dnd MO in various species already (zebrafish^{1,2}, goldfish³, sturgeon⁴). Therefore we did not paid full attention to it. Here MO based strategy was rather chosen as the most rapid and straightforward way how to obtain suitable recipients for transplantation. We are aware that MO injection is hand to hand with the appearance of typical phenotypes such as heart edema, which we also observed in other species. However, we believe that this is rather general effect of the MO toxicity, not the off-target effect. Also, we did not intend to thoroughly investigate the embryonic or larval phenotypes of dnd morphants. The rescue experiment combining MO and dnd mRNA would also be difficult because we do not know full sequence of the giant danio dead end gene.

Addressing the concerns of specificity, we are providing data from coinjection of dnd MO with gfpUTRnanos3 where we showed depletion of the primordial germ cells. Later, we performed histology and immunohistochemistry with ddx4 antibodies to confirm the depletion of germ cells in the gonads of adult fish. Undoubtedly, precise identification of the potential off-target effect would strengthen the used methodology; however, we are not sure whether it would change the results and conclusion of the study. Our opinion is that the timing of germ cell transplantation which is done after swim up stage at the start of exogenous feeding is actually very convenient because it works as the selection barrier when fish with defective phenotypes cannot either hatch or reach swim up stage properly. Therefore, only “healthy” and viable larvae are later used for the experiment.

5. This manuscript also missed some highly relative literatures, such as zebrafish were used as surrogate host to produce sperm of other species from another subfamily by spermatogonial cell transplantation (<https://doi.org/10.1007/s11427-021-1989-9>).

We apologize for missing this literature and thank the reviewer for bringing this to our notice. We have cited this reference in the manuscript (reference 52, line 493).

6. All the transgenic fish and gene name should follow the standard nomenclature of zebrafish, such as Tg(ddx4:egfp), instead of vas::EGFP in the manuscript, gfpUTRnanos3, instead of GFP-nos1-3'UTR in the manuscript.

We apologize for not providing the standard names for the genes. We have modified vas::EGFP to Tg(ddx4:egfp) and GFP-nos1-3'UTR to gfpUTRnanos3 in the whole manuscript.

Minor points:

1. Line 29, should clearly define the genetic relationship between zebrafish and giant danio.

Corrected. Now line 29 is changed to “closely related larger species from the same subfamily, giant danio *Devario aequipinnatus*”.

2. Line 59, it is not clear whether they used germ stem cells for transplantation, thus it is not appropriate to use the phrase “germ stem cells” or “spermatogonial stem cells” in this study. Suggested to use the phrase “spermatogonial cells” in the whole manuscript.

We thank the reviewer for this suggestion. Now “germ stem cells” and “spermatogonial stem cells” is changed to “spermatogonial cells” in the whole manuscript.

3. Line 63, should give a reference for the maturation period of giant danio. Otherwise, this description should be moved to the Results section.

This description is now deleted and moved to the result section 3.2) Transplant success; germline chimera reproduction, line 356.

4. Line 66-68, should provide reference for this sentence. Otherwise, this description should be moved to the Results section.

The number of eggs and milt volume is moved to the result section 3.1 “Recipient production” Line 280.

5. P70, an extra dot.

Corrected.

6. P86, are > were.

Corrected.

7. Line 97-98, this conclusion is not accurate. It is possible to obtain donor-derived oocytes when zebrafish PGCs were transplanted into germ cell-depleted host embryos (<https://doi.org/10.1073/pnas.222459999>, <https://doi.org/10.1016/j.jgg.2019.12.004>).

We apologize for this statement. These authors have obtained donor-derived oocytes by PGC transplantation into the host embryos at the blastula stage and a high amount of PGCs should be transplanted to produce female germline chimera in zebrafish. However, it is impossible to generate female chimera by spermatogonial cell transplantation into the larvae. We have corrected the line 92-95 as follows.

“Many studies have shown that zebrafish develop male-like characteristics after endogenous germ cell depletion making it impossible to obtain donor-derived oocytes through sterile zebrafish surrogates by spermatogonial cell transplantation into the larvae”.

8. Figure 1, it is not accurate to use the phrase “spermatogonial stem cells”, and it is also needed to clearly show how to obtain the donor cells for transplantation.

Figure 1 is modified with three additional steps showing the enzymatic dissociation of testicular tissue followed by density gradient centrifugation with Ficoll-Paque™ PLUS.

(Corrected schematic diagram for figure 1)

9. Line 114, there should be a reference for the Tg(ddx4:egfp) transgenic line.

Reference is now cited.

10. Line 123-125, it is not clear why the primers for amplifying giant danio dnd gene was designed according to the zebrafish dnd1 mRNA sequence.

We apologize for this confusion. Dead end is an evolutionarily conserved gene among vertebrates^{5,6}, and giant danio is a closely related species to zebrafish. The MO is designed to target the post-spliced mRNA and block the translation of the dead end gene. This approach is helpful when the complete genome sequence of a species is unknown⁷, and in our study, we do not know the genome sequence of giant danio. Thus, we have amplified the giant danio dead end cDNA with zebrafish dnd1 mRNA primers. We also performed nucleotide BLAST for the amplified sequence, and the result was 89% matching with the zebrafish dnd and 87% identical

to dnd of other fish species. MO sequence was designed according to the guidelines of GeneTools, Philomath, OR.

Description	Scientific Name	Max Score	Total Score	Query Cover	E value	Per. Ident	Acc. Len	Accession
Danio rerio dead end mRNA (cDNA clone MGC:191688 IMAGE:100059997) complete cds	Danio rerio	394	394	83%	6e-105	89.87%	1303	BC164513.1
Danio rerio DND microRNA-mediated repression inhibitor 1 (dnd1) mRNA	Danio rerio	394	394	92%	6e-105	87.50%	1765	NM_212795.1
Danio rerio dead end mRNA (cDNA clone MGC:111864 IMAGE:7412116) complete cds	Danio rerio	394	394	92%	6e-105	87.50%	1839	BC095578.1
PREDICTED: Sinocyclocheilus grahami dead end protein 1-like (LOC107587089) mRNA	Sinocyclocheilu...	366	366	82%	1e-96	88.52%	2143	XM_016274729.1
PREDICTED: Sinocyclocheilus rhinoceros dead end protein 1-like (LOC107731723) mRNA	Sinocyclocheilu...	363	363	80%	2e-95	88.89%	1598	XM_016542874.1
PREDICTED: Sinocyclocheilus grahami dead end protein 1-like (LOC107559292) mRNA	Sinocyclocheilu...	363	363	80%	2e-95	88.89%	2094	XM_016243058.1
PREDICTED: Sinocyclocheilus rhinoceros dead end protein 1-like (LOC107758536) mRNA	Sinocyclocheilu...	361	361	86%	6e-95	87.46%	2042	XM_016576295.1
PREDICTED: Ctenopharyngodon idella DND microRNA-mediated repression inhibitor 1 (LOC127494684) mRNA	Ctenopharyngop...	351	351	81%	4e-92	88.04%	2374	XM_051860768.1
Rhodeus ocellatus ocellatus dead end mRNA complete cds	Rhodeus ocellat...	342	342	80%	2e-89	87.54%	1613	MG995743.1
PREDICTED: Carassius auratus dead end protein 1-like (LOC113113732) mRNA	Carassius auratus	340	340	80%	8e-89	87.54%	1857	XM_026280173.1
Carassius gibelio dead end (DND) mRNA complete cds	Carassius gibelio	340	340	80%	8e-89	87.54%	1499	KP641680.1
PREDICTED: Megalobrama amblycephala DND microRNA-mediated repression inhibitor 1 (dnd1) trans...	Megalobrama a...	340	340	81%	8e-89	87.38%	1665	XM_048176494.1
PREDICTED: Megalobrama amblycephala DND microRNA-mediated repression inhibitor 1 (dnd1) trans...	Megalobrama a...	340	340	81%	8e-89	87.38%	1746	XM_048176493.1
PREDICTED: Cyprinus carpio dead end protein 1-like (LOC109089193) transcript variant X1 mRNA	Cyprinus carpio	340	340	80%	8e-89	87.46%	1790	XM_042770657.1
Cyprinus carpio dead end (DND) mRNA complete cds	Cyprinus carpio	340	340	80%	8e-89	87.46%	1128	MN447719.1

(Image of dnd gene BLAST)

11. Line 139-140, there should be an approach to clearly define the specificity and effectiveness of the newly designed MO.

As explained above, we have co-injected morpholino with gfpUTRnanos3 to confirm the specificity of the designed MO. The function of nos1 3'UTR is widely conserved across fish species⁸, and we separately injected one group (n = 50) with only gfpUTRnanos3 and observed strong GFP expression in the PGCs during the segmentation period (10-24 hour post-fertilization), but the MO plus gfpUTRnanos3 group (n = 50) did not show GFP expression in any of the embryos.

12. Line 150, GFP-nos1-3'UTR should be gfpUTRnanos3.

We thank the reviewer for pointing this out. GFP-nos1-3'UTR is now changed to gfpUTRnanos3 in the whole manuscript.

13. Line 160, swim-up stage should be standard stage name, according to https://zfin.org/zf_info/zfbook/stages/index.html. There is an error for the reference.

We apologize for not mentioning the standard stage names. Swim-up started on day 5 post-fertilization and the stage name is now changed to the standard name in the whole manuscript.

This error was for 'table 1' citation. We have corrected this.

14. Line 163, delete the word "fish".

We are sorry for this error, now it is corrected.

15. Line 254, please provide the standard name of the gene "elongation factor".

The standard name for the reference gene eukaryotic translation elongation factor 1 alpha 1, like 1 is now provided, which is *eef1a1l1*.

16. Line 282, define PGCs here.

We have defined the PGCs as follows (Line 285-287)

PGCs are the precursors of germ cell lineage that arise at the marginal part of blastodisc at the blastula stage and migrate to the genital anlage during embryogenesis

17. Line 284, lack statistic data. There is an error for the reference,

We are sorry for not providing the data for PGC depletion efficiency. We injected 50 embryos with gfpUTRnanos3 (positive control to label the PGCs) at 1-2 cell stage to the giant danio recipients and another group with 'MO plus gfpUTRnanos3'. A total of 50 embryos were injected for both groups. We could not observe GFP expression in any MO plus gfpUTRnanos3 embryos, whereas the positive control groups showed prominent GFP expression for labeled PGC As shown in 2a-b'. The efficiency was 100% according to these results. Line 287-292.

We are sorry for the error. The reference figure 2a-b' is cited again (line 290).

18. Figure 2g, please compare the expression levels of each gene between different samples, but not different genes in a certain sample.

We thank the reviewer for this suggestion. Now we have compared the expression levels of each gene between different samples and the graph is modified as below (Figure 2f, the graph represents mean \pm SD).

19. Line 322, please use the standard stage names for these stages in Table 1.

We are sorry for not following the standard names. It is corrected in Table 1.

20. Line 325, please remove all the “AB” in Table 2. It is unnecessary to mention AB in the Table.

AB is now removed in Table 2 and Table 3.

21. Line 357, Figure 3, please label the genotype or injection condition in the panels.

Figure 3 describes the result of Vasa-immunohistochemistry in sterile giant danio and controls. This figure is now merged in Figure 2 (g-l’). We have not performed any injections here.

22. Line 396-308, needs to verify the germline-specific EGFP signal in the F2 progeny.

Germline-specific GFP expression is now checked, and the images are provided in Figure 4g-l’ and cited in the text (line 414)

23. Line 401, Table 3, how many replicates have been conducted to test the fertilization rate? How could the authors obtain exactly 100 eggs in each group?

Five biological replicates (one male = one biological replicate, one group has 5 males) were conducted for the fertilization trial to determine the fertilization rate and the survival rate until day-5 for each group (source data is in supplementary file 2). Fertilization trials were repeated several times with the germline chimera males and the controls during the germline chimera reproduction.

For in-vitro fertilization, eggs were collected from the females and pooled, then divided according to five replicates for each group. Immediately after fertilization, 100 eggs were randomly transferred to a new Petri dish for further counting. Survival was recorded from the 256-blastula stage until day 5.

24. Line 426, 104 µl should be corrected.

We are sorry for this error. Now it is corrected 10⁴ µl.

25. Line 433, Figure 7, the author should check the germline-specific expression of EGFP in the F2 progeny.

Germline-specific GFP expression is now checked, and the images are provided in Figure 4g-l'.

26. Line 461-462, it is true that EGFP has a maternal contribution, but the authors need to check the germline expression at later stages.

Germline-specific GFP expression is now checked, and the images are provided in Figure 4g-l'.

27. Line 477-478, it is not necessarily true. In previous studies, germ-cell transplanted zebrafish also reached sexual maturation at 3 months ().

We agree with the reviewer that some zebrafish chimera reached sexual maturity at three months post-transplantation. However, this scenario is not always the same. Marinović et al.⁹ and Wong et al.¹⁰ obtained donor-derived gametes six months post-transplant in zebrafish. We have now modified the text (line 473-475) as follows

“This observation is not new, as few previous studies have reported zebrafish to become sexually mature ~ 6 months after germ cell transplantation.”

28. Line 480, ASG should be SPG-A.

ASG is corrected to SPG-A in the whole manuscript.

29. Line 490, *cyp191a* should be *cyp19a1a*

We are sorry for this error. Now it is corrected to *cyp19a1a*

30. Line 491, secretion should be production.

Corrected (line 487)

31. Line 496, a reference is missing (<https://doi.org/10.1007/s10126-019-09874-1>).

This reference is now cited, reference number 50 (Line 492)

32. Line 497, spermatogonial cell transplantation from another species into zebrafish was also achieved (<https://doi.org/10.1007/s11427-021-1989-9>)

We are sorry for not citing this paper. We have now cited this reference (reference 52, line 493)

Reviewer #2 (Remarks to the Author):

Zebrafish is a useful experimental model, but its low semen volume makes sperm cryopreservation difficult. The authors have succeeded in producing sperm derived from a donor zebrafish from 6-month-old recipients by preparing germline stem cells from the testis of a 3-month-old zebrafish and transplanting them into the recipients. The authors estimate that the amount of sperm produced by the recipient was 10 times more than that of normal zebrafish. Although this idea is very interesting, this reviewer did not find sufficient merit in this technique for the following two reasons.

(1) Zebrafish biobanking usually involves mincing the testes to obtain a large amount of sperm, making it possible to obtain a sufficient amount of samples for cryopreservation.

We agree with the reviewer that minced zebrafish testes can generate a large amount of sperm for cryopreservation. However, we have aimed at many other applications in our study, such as DNA/RNA extraction and in-vitro fertilization from one individual. Some experimental designs require dividing the semen samples into many replicates, which seems difficult with 1µl sperm from zebrafish males. Our idea is not to replace the already established methods for cryopreservation but to provide an alternative that may tackle problems associated with zebrafish's low-milt production.

(2) In this study, 3-month-old individuals were used as donors, and a large amount of zebrafish sperm was obtained 6 months after the germ cell transplantation. With this amount of time, two more generations of zebrafish can be mated, which can increase the number of sperm-producing individuals by several hundred times.

The resulting germline chimera in our study can produce ten times more concentrated cells in a single spawning. As we explained above, this amount of sperm is enough to conduct several downstream experiments from one single male. We agree with the reviewer that zebrafish can be bred in two generations to get more sperm, but also, we need to consider some experimental designs that need samples from one individual and cannot wait for their second generation to get sexually mature in order to obtain the milt. Moreover, this case is not ideal when one wishes not to pool the samples from different males from different generations.

In addition, this phenomenon is not new from a biological point of view, since germ cell transplantation using zebrafish as donors has already been reported in several papers, and interspecific germ cell transplantation within the same genus has also been reported successfully in several fish species.

We agree with the reviewer that zebrafish has been used as a donor for many germ cell transplantation studies. However, zebrafish spermatogonia cells have never been transplanted into a species from a different genus with viable donor-derived sperm production or to increase sperm production to the best of our knowledge.

Reviewer #3 (Remarks to the Author):

General comment:

This manuscript reported the germ-line chimeras from zebrafish as donor to giant danio as host, in which sperm derived from donor were produced. In the germ-line chimera, volume and concentration of sperm were higher than those of donor species. As mentioned in the manuscript by the authors, this finding would be able to contribute several applications such as sperm cryopreservation. From another view point, this result is interesting to study the definition of gonadal size in species. But the authors did not mention the size of testes of germ-line chimeras and compared size of testes among germ-line chimera and species used in this study. Furthermore, it might be possible to discuss the ability of proliferation in germ cells derived from donor species. I wonder if the authors could add the information about the size of their gonads and discuss the gonadal development of germ-line chimera to improve this manuscript for basic biological interests.

We thank the reviewer for the positive remarks on the manuscript. The gonadal size comparison is a very nice idea, and we thank the reviewer for this. We will consider it in future studies.

Specific comments:

Line 160. No need to cite Table 1 in this part.

We thank the reviewer for this suggestion. Citation is now removed.

Line 199-210. How did you conduct artificial fertilization? How many eggs and how much volume of diluted semen were used for the fertilization trials in this study, and then how did you activate them for fertilization?

We apologize for not providing enough details for in-vitro fertilization and we have modified the texts as follows (Line 199 – 207).

“On the spawning day, barriers were removed, and the females were observed for oviposition. Breeding pairs were first anesthetized with 0.08% MS222. Eggs from 7-8 zebrafish ovulated females were collected together in one Petri dish by gentle abdominal pressure. Similarly, two ovulated giant danio females were stripped to obtain the eggs. Semen samples from the control males and the germline chimera males were collected separately in the E400 extender (10 µl of the extender was used for each 1 µl of milt). 10-20 µl of diluted sperm was used to fertilize ~ 100 eggs for each group (0.2 -0.5 ml of dechlorinated water was used for sperm activation).”

Line 350-352. Figure 5e should be cited.

We thank the reviewer for pointing this out. Figure 5e which is now changed to Figure 3j (After the combinations suggested by Reviewer#1) is cited in the text (Line 359).

Line 353-356. In figure 5d-d', the authors indicated four spermatozoa by arrowheads. Could you detect GFP expression in all sperm observed

The arrowheads are only used to indicate the GFP expression near the mid-piece of spermatozoa, but the expression was observed in all sperm cells.

Reference

1. Franěk, R. *et al.* Who is the best surrogate for germ stem cell transplantation in fish? *Aquaculture* **549**, 737759 (2022).
2. Saito, T., Goto-Kazeto, R., Arai, K. & Yamaha, E. Xenogenesis in teleost fish through generation of germ-line chimeras by single primordial germ cell transplantation. *Biol. Reprod.* **78**, 159–166 (2008).
3. Franěk, R., Kašpar, V., Shah, M. A., Gela, D. & Pšenička, M. Production of common carp donor-derived offspring from goldfish surrogate broodstock. *Aquaculture* **534**, (2021).
4. Linhartová, Z. *et al.* Sterilization of sterlet *Acipenser ruthenus* by using knockdown agent, antisense morpholino oligonucleotide, against dead end gene. *Theriogenology* **84**, 1246-1255.e1 (2015).
5. Baloch, A. R., Franěk, R., Saito, T. & Pšenička, M. Dead-end (dnd) protein in fish—a review. *Fish Physiol. Biochem.* **47**, 777–784 (2021).
6. Weidinger, G. *et al.* dead end, a Novel Vertebrate Germ Plasm Component, Is Required for Zebrafish Primordial Germ Cell Migration and Survival. *Curr. Biol.* **13**, 1429–1434 (2003).
7. Eisen, J. S. & Smith, J. C. Controlling morpholino experiments: don't stop making antisense. *Development* **135**, 1735–1743 (2008).
8. Maegawa, S. *et al.* Visualization of primordial germ cells in vivo using GFP-nos1 3'UTR mRNA. *Int. J. Dev. Biol.* **50**, 691–700 (2002).
9. Marinović, Z. *et al.* Preservation of zebrafish genetic resources through testis cryopreservation and spermatogonia transplantation. *Sci. Reports* **2019 91 9**, 1–10 (2019).
10. Wong, T. T., Saito, T., Crodian, J. & Collodi, P. Zebrafish germline chimeras produced by transplantation of ovarian germ cells into sterile host larvae. *Biol. Reprod.* **84**, 1190–1197 (2011).

REVIEWERS' COMMENTS:

Reviewer #1 (Remarks to the Author):

The reviewer is satisfied with the current revision of the manuscript.

Reviewer #2 (Remarks to the Author):

The revised manuscript is well written and the content should be of interest to the zebrafish community.

Reviewer #3 (Remarks to the Author):

The authors' answer to the general comment: We thank the reviewer for the positive remarks on the manuscript. The gonadal size comparison is a very nice idea, and we thank the reviewer for this. We will consider it in future studies.

Comment to the answer: In the revised manuscript, the difference of gonadal size between zebrafish and giant danio is not mentioned. In the discussion, the authors need to mention the possible reason why the milt volume was increased in the germline chimera. It would be of value to the reader to indicate the biological interest in this study.

Point-by-point response to the referees

REVIEWERS' COMMENTS:

Reviewer #1 (Remarks to the Author):

The reviewer is satisfied with the current revision of the manuscript.

We thank the reviewer for a detailed review. All the comments have helped us to improve the manuscript.

Reviewer #2 (Remarks to the Author):

The revised manuscript is well written and the content should be of interest to the zebrafish community.

We thank the reviewer for the positive comments on the manuscript.

Reviewer #3 (Remarks to the Author):

The authors' answer to the general comment: We thank the reviewer for the positive remarks on the manuscript. The gonadal size comparison is a very nice idea, and we thank the reviewer for this. We will consider it in future studies.

Comment to the answer: In the revised manuscript, the difference of gonadal size between zebrafish and giant danio is not mentioned. In the discussion, the authors need to mention the possible reason why the milt volume was increased in the germline chimera. It would be of value to the reader to indicate the biological interest in this study.

We thank the reviewer for the suggestion. Now we have included the following text in the discussion (line 330-335)

“A significant increase in sperm volume and concentration in giant danio surrogates explains the relationship between body mass and reproductive output. Reportedly, reproduction output increases isometrically with body weight and size⁵⁰. Gamete biomass scaled sub-linearly with